# Peripheralization Strategies Applied to Morphinans and Implications for Improved Treatment of Pain

**DOI:** 10.3390/molecules28124761

**Published:** 2023-06-14

**Authors:** Helmut Schmidhammer, Mahmoud Al-Khrasani, Susanna Fürst, Mariana Spetea

**Affiliations:** 1Department of Pharmaceutical Chemistry, Institute of Pharmacy and Center for Molecular Biosciences (CMBI), University of Innsbruck, Innrain 80-82, 6020 Innsbruck, Austria; helmut.schmidhammer@uibk.ac.at; 2Department of Pharmacology and Pharmacotherapy, Faculty of Medicine, Semmelweis University, Nagyvárad tér 4, H-1445 Budapest, Hungary; al-khrasani.mahmoud@med.semmelweis-univ.hu (M.A.-K.); furst.zsuzsanna@med.semmelweis-univ.hu (S.F.)

**Keywords:** pain, analgesia, opioid receptors, peripheral analgesia, peripherally restricted opioids, morphine, morphinans, drug design strategies, structure–activity relationships, opioid side effects

## Abstract

Opioids are considered the most effective analgesics for the treatment of moderate to severe acute and chronic pain. However, the inadequate benefit/risk ratio of currently available opioids, together with the current ‘opioid crisis’, warrant consideration on new opioid analgesic discovery strategies. Targeting peripheral opioid receptors as effective means of treating pain and avoiding the centrally mediated side effects represents a research area of substantial and continuous attention. Among clinically used analgesics, opioids from the class of morphinans (i.e., morphine and structurally related analogues) are of utmost clinical importance as analgesic drugs activating the mu-opioid receptor. In this review, we focus on peripheralization strategies applied to *N*-methylmorphinans to limit their ability to cross the blood–brain barrier, thus minimizing central exposure and the associated undesired side effects. Chemical modifications to the morphinan scaffold to increase hydrophilicity of known and new opioids, and nanocarrier-based approaches to selectively deliver opioids, such as morphine, to the peripheral tissue are discussed. The preclinical and clinical research activities have allowed for the characterization of a variety of compounds that show low central nervous system penetration, and therefore an improved side effect profile, yet maintaining the desired opioid-related antinociceptive activity. Such peripheral opioid analgesics may represent alternatives to presently available drugs for an efficient and safer pain therapy.

## 1. Introduction

Effective and adequate management of pain, particularly chronic pain, is still an area of unmet medical need at the beginning of the third millennium. Opioids are the gold standard for the treatment of moderate to severe acute and chronic pain [1]. However, strong opioid analgesics, such as morphine, oxycodone and fentanyl, are not effective for pain relief in all patients, nor are they well-tolerated by all patients, because of an array of severe side effects, including respiratory depression, constipation, sedation, nausea and dizziness [2]. With prolonged use, opioid safety is dramatically reduced because of analgesic tolerance, and risks of physical dependence and addiction, promoting the development of opioid use disorders and overdose deaths [2]. Moreover, the misuse of prescription opioids (i.e., oxycodone, hydrocodone and fentanyl) [3], as well as over-the-counter opioids (i.e., codeine, hydrocodeine and loperamide) [4], has led to the current opioid epidemic, particularly in North America and Europe. In the USA, the opioid-involved overdose deaths had increased to 80,411 in 2021 [5]. The recent explosion in fatalities resulting from overdose of prescription and synthetic opioids, especially fentanyl and its various analogs [3,5], has dramatically increased the need for safer analgesics.

Opioid receptors represent the most important players in pain modulation and are the molecular targets of clinically used opioids. There are three main opioid receptor types, mu (MOR), delta (DOR) and kappa (KOR), and the non-classical receptor, nociceptin/orphanin FQ (NOP) receptor [1]. All opioid receptor types belong to the family of G protein-coupled receptors (GPCRs) with seven transmembrane domains, and are expressed throughout the central and peripheral nervous systems (CNS and PNS, respectively), and in various non-neuronal tissues [1,6,7]. Because of their therapeutic relevance, opioid receptors are among the few GPCRs determined in different activation states, providing important information on the type-specific binding characteristics of ligands [8]. Although opioid receptors contribute to pain inhibition, the MOR is recognized as the dominant type for its pain-relieving effects [2,9,10]. The major drawback of targeting the MOR for clinical analgesia is that it is also responsible for the undesirable side effects. Most of the detrimental side effects, including respiratory depression, sedation, analgesic tolerance, reward and dependence, are mediated by the MOR in the CNS, whereas constipation is mainly mediated by the MOR in the intestinal myenteric plexus [2,11,12].

Among therapeutically valuable opioids, morphinans are of the utmost clinical importance as analgesic drugs because of their agonistic actions to the MOR. They include powerful pain relieving agents, such as naturally occurring alkaloids (e.g., morphine and codeine), semisynthetic analogues (e.g., hydrocodone, hydromorphone, oxycodone, oxymorphone and buprenorphine), and synthetic derivatives (e.g., levorphanol) (Figure 1) [9,13,14,15]. Morphine and structurally related MOR agonists also share similar side effects, including addictive liability. Because of their outstanding medicinal relevance in combination with an attractive chemical scaffold, morphinan alkaloids represent attractive synthetic targets. Therefore, diverse research approaches toward morphine and its congeners have been devised for mitigating their deleterious side effects and limiting abuse and misuse (for reviews, see [9,13,14,15,16,17,18,19,20]).

Experimental and clinical studies provide substantial evidence that opioid analgesia is not exclusively mediated via the central opioid receptors (for reviews, see [21,22,23,24]). Pharmacological, neuroanatomical, molecular and electrophysiological studies have shown that peripheral opioid receptors are expressed on the peripheral terminals of sensory neurons, where they can modulate both afferent and efferent neuronal functions, resulting in potent and clinically measurable analgesia. Early clinical studies using intra-articular morphine administration in conjunction with arthroscopy in the knee joint supported the notion that the activation of peripheral opioid receptors induced pain relief by a peripheral mechanism and did so without side effects [25,26]. These findings have led to new research directions aiming on targeting the peripheral opioid receptors for superior pain management.

From a medicinal chemistry point of view, and particularly for the clinical implication to improve pain treatment, considerable effort has been directed towards the peripheralization of opioids with the objective of minimizing central exposure and their ability to penetrate the blood–brain barrier (BBB) and thus the related undesirable side effects (Figure 2). In this review, we focus on peripheralization strategies applied to the morphinan class of opioids for creating effective and safer medications for pain.

## 2. Peripheralization Strategies Applied to Morphinans

Different chemical strategies have been developed to limit the ability of opioids to cross the BBB, including (a) chemical modifications to the morphinan skeleton to increase hydrophilicity of known and new opioids, and (b) nanocarrier-based approaches to selectively deliver opioids, such as morphine to the peripheral tissue. The following sections discuss significant representatives, including design strategies, synthetical procedures, pharmacology and structure–activity relationships (SAR).

### 2.1. Quaternization of the Morphinan Nitrogen

The first effort to minimize the CNS effects of opioids while retaining their actions in peripheral tissue was the quaternization of the nitrogen in the clinically used morphine, oxymorphone, nalorphine, naloxone and naltrexone (Figure 3) [27,28]. Peripheral selectivity of the quaternary derivative of morphine, *N*-methylmorphine, was described over 50 years ago [29]. The systemic intravenous (i.v.) administration of *N*-methylmorphine caused the inhibition of gastrointestinal transit because of its action on the opioid receptors in the gut. In the hot-plate test, centrally mediated antinociception was produced by 15 mg/kg morphine in mice after intraperitoneal (i.p.) administration but not by *N*-methylmorphine at the same dose [29]. Furthermore, *N*-methylmorphine proved to be ineffective in the hot-plate test even in a dose 100 mg/kg [30]. In an acetic-acid-induced writhing assay, *N*-methylmorphine produced antinociceptive effects in mice after i.p. administration in a dose of 45 mg/kg, being 30-fold less potent than morphine [30]. *N*-methylmorphine was also shown to selectively inhibited phase II in the formalin test following systemic i.p. administration [31]. The antinociceptive effect of *N*-methylmorphine after central intracerebroventricular (i.c.v.) administration was antagonized by systemically applied naloxone but not by peripheral antagonist *N*-methylnaloxone, showing the peripheral site of action of *N*-methylmorphine [31].

It is important to note that quaternization of nitrogen on morphine-based structure derivatives has been reported to have negative impact on both the affinity to the opioid receptor and the agonist activity of generated analogues [28,32]. Quaternization also trends to reduce potency in vivo. Therefore, alternative strategies to limit BBB penetration have been pursued.

### 2.2. Introduction of Hydrophilic Substituents at Position 6

Polar or ionizable substitutions are able to increase polarity and inhibit the crossing of the BBB. Therefore, opioids with hydrophilic groups attached to the C-6 position of the morphinan skeleton were designed. The first examples of morphinans having ionizable residues at position 6 were reported more than 30 years ago. They were synthesized from β-oxymorphamine [33], β-naltrexamine [33] and β-funaltrexamine [34]. Such compounds with zwitterionic moieties showed significantly reduced access to the CNS without substantially decreased opioid receptor in vitro and in vivo activity [33,34].

Noteworthy are the 6-amide derivatives of β-oxymorphamine (**a**–**e**, Figure 4) reported as the first peripherally selective opioid agonists and effective antinociceptives [33]. All compounds have C-6 moieties that are ionized at the pH of the gut or at physiologic pH, accounting for a more restricted capability to enter the CNS than the unionized molecules. Compounds **a**, **c** and **d** were synthesized from β-oxymorphamine with the appropriate anhydride [33]. The fumaramic acid **b** was prepared by coupling the half-ester of fumaric acid with β-oxymorphamine and then subjecting the fumaramate esters to hydrolysis. The aspartyl derivative **e** was obtained through coupling BocAsp γ-benzyl ester with β-oxymorphamine, followed by deprotection with acid to remove the Boc group and hydrogenolysis of the benzyl function [33]. As regards biological activities, the ß-oxymorphamine derivatives **a**–**e** were all full agonists in the guinea pig ileum (GPI) bioassay with potencies that were 1.5- to 6-fold higher than the potency of morphine (Table 1) [33]. In a mouse model of acute thermal nociception, the tail-flick assay, all compounds possessed potent antinociceptive activity when administered by the i.c.v route (Table 1). They also were active in inducing antinociception when given systemically by i.v. administration to mice. When compared on a body weight basis, the i.v. ED_50_ doses were about 1000-fold higher than the i.c.v. ED_50_ values. Derivatives **a** and **c** were also active when given orally (p.o.) (Table 1) [33]. The attachment of polar groups, particularly zwitterionic moieties, at the C-6 position of the morphinan structure is effective in excluding such ligands from the CNS, thereby affording peripheral selectivity.

Other more recent examples of morphinans with ionizable groups at position 6 emerging as peripheral opioid antinociceptives with restricted penetration into the CNS are described in the following sections.

#### 2.2.1. 6-Amino-acid-substituted 14-Alkoxymorphinans

Our first synthetic efforts directed towards the development of ionizable molecules in the class of 14-alkoxymorphinans as peripherally acting opioid analgesics started with the series of six 6-amino acids, i.e., Gly-, L-Ala- and L-Phe- substituted derivatives; **2a/b** (HS-730/HS-731); **3a/b** (HS-935/HS-936); and **4a/b** (HS-937/HS-938), respectively, of the highly potent and centrally acting MOR agonist 14-*O*-methyloxymorphone (14-OMO, **1**) (Figure 1) [35]. A novel synthetic procedure for the synthesis of 6-amino-acid-substituted derivatives in the morphinan series was used. The *tert*-butyl ester derivatives **2aa/bb**, **3aa/bb**, and **4aa/bb** were prepared from 14-OMO (**1**) by reductive amination with the respective *tert*-butyl ester hydrochlorides and sodium cyanoborohydride in ethanol. After separating the diastereoisomers by column chromatography, esters **2aa/bb**, **3aa/bb**, and **4aa/bb** were treated with tetrafluoroboric acid in dichloromethane to afford 6-Gly (**2a** and **2b**), 6-Ala (**3a** and **3b**) and 6-Phe (**4a** and **4b**) substituted derivatives, respectively (Figure 1) [35].

We further targeted derivatization of 14-OMO (**1**) through introduction of other amino acid residues of the L- and/or D-series at position 6, including natural amino acids, i.e., Ser, Val, Lys, Tyr, Trp, Asn, Gln, Asp and Glu (**5a**–**13b**, Figure 2), and unnatural amino acids, i.e., D-Ala, D-Val, D-Phe, L-Chg (L-cyclohexylglycine), L-Abu (L-2-aminobutyric acid), β-Ala and GABA (γ-aminobutyric acid) (**14a**–**20b**, Figure 2) [36]. Additionally, three zwitterionic molecules with a dipeptide substitution at position 6, i.e., L-Val-L-Tyr and Gly-Gly in 14-OMO (**21a**–**22a**, Figure 3) were synthesized [36]. The reductive amination of 14-OMO (**1**) was performed using amino acid *tert*-butyl ester hydrochlorides or dipeptide benzyl ester hydrochlorides, and NaBH_3_CN in CH_3_OH. Medium-pressure liquid chromatography (MPLC) was used to separate the diastereoisomers, providing ester derivatives **5aa**–**22aa** (Figure 2 and Figure 3). Typically, the ratio of 6β-amino to 6α amino epimers was between 4:1 and 2:1. The 6-amino-acid (**5a**–**20b**)-substituted derivatives were obtained through ester cleavage of the *tert*-butyl derivatives in dioxane/HCl (Figure 2). Catalytic hydrogenation of the benzyl esters **21aa/bb** and **22aa** in CH_3_OH using 10% Pd/C catalyst provided the 6-dipeptide-substituted **21a/b** and **22a**, respectively (Figure 3) [36].

Synthetic work also targeted the combination of 6-amino amino (i.e., Gly) and 14-arylalkoxy (e.g., phenylpropoxy) substitutions in *N*-methyl-morphinans (Figure 4) [37]. The reductive amination of the 14-phenylpropoxyoxymorphone (POMO, **23**) was performed with Gly *tert*-butyl ester hydrochloride and NaCNBH_3_ in DMF/MeOH at room temperature. The diastereoisomers were separated by column chromatography to obtain *tert*-butyl esters **24aa** and **24bb**. Ester cleavage in dioxane/HCl generated the amino acids **24a** and **24b** (Figure 4) [37].

SAR studies on the series of amino acid and dipeptide substitution at position 6 in 14-OMO (1) as zwitterionic molecules explored their binding and activation of the opioid receptors and antinociceptive properties (Table 2, Table 3 and Table 4). The 6-amino acid groups included natural amino acids (i.e., Gly, Ala, Phe, Ser, Val, Lys, Tyr, Trp, Asn, Gln, Asp and Glu), unnatural amino acids (i.e., D-Ala, D-Val, D-Phe, L-Chg, L-Abu, β-Ala and GABA) (**2a**–**20b**) (Figure 1, Figure 2 and Figure 4) [35,36,37,38], and 6-dipeptide substitution (i.e., L-Val-L-Tyr and Gly-Gly) (**22a/b** and **23a**) (Figure 3) [36]. In vitro receptor binding (radioligand binding assays with membranes from rodent brain and CHO cells expressing the human opioid receptors) and functional assays (mouse vas deferens (MVD) bioassay and [^35^S]GTPγS binding assay with membranes from CHO cells expressing the human opioid receptors) established the potent MOR/DOR agonist profile and reduced the binding and activation of the KOR for most compounds (Table 2). The replacement of the 14-methoxy group in *N*-methyl, 6-Gly substituted morphinans **2a** (HS-730) and **2b** (HS-731) with a 14-phenylpropoxy group (compounds **24a** and **24b**, respectively) resulted in a considerable increase in binding affinities to all three opioid receptor types, MOR, DOR and KOR, in rodent brain membranes (Table 2 and Table 3) [37]. Compared to the nonselective 14-phenylpropoxy-substituted POMO (**23**) [39], the 6-Gly analogues **24a** and **24b** showed a comparable binding profile to the opioid receptors, acting as mixed MOR/DOR/KOR ligands (Table 2) [37].

Derivatives with unnatural amino acids at position 6 generally presented high MOR and DOR binding affinities and agonism, similar to compounds with natural amino acids (Table 2). Substituting L-amino acids by D-amino acids left MOR binding affinities and agonist potencies unchanged or caused an increase but the MOR full agonism was largely retained (Table 2) [36]. The full agonism to the DOR was not affected by the replacement of L-amino acids with D-amino acids, while conversion from partial agonists to full agonists at KOR was observed. While the α-epimers were frequently favored for the MOR by strongly activating this receptor, the β-epimers showed increased binding and potent activation of the DOR (Table 2). Opioid activity of 6-dipeptide-substituted derivatives was also reported, with analogues **22a** and **22b** as highly potent MOR partial agonists and very potent DOR full agonists, while 23a was less potent and a MOR/DOR full agonist (Table 2) [36].

In vivo, the 6-amino-acid- and 6-dipeptide-substituted *N*-methymorphinans were reported as very effective in inducing antinociceptive effects in rodent pain models of acute pain, visceral pain, inflammatory pain and trigeminal nociception after systemic (s.c., i.p. and p.o.) administration (Table 4) [36,37,40,41,42,43]. In the radiant heat tail-flick test in rats, they were up to 200-fold more potent than morphine after s.c. administration, and had similar potencies to fentanyl, with markedly longer duration of action (Table 4) [37,40]. A similar profile was reported following central i.c.v. administration of the 6-amino acid conjugates of 14-OMO (**1**), i.e., **2a/b** (HS-730/HS-731), **3a/b** (HS-935/HS-936) and **4a/b** (HS-937/HS-938) [40].

All compounds, except 6β-L-Phe substituted **4b** (HS-938), were more effective in producing an antinociceptive response than morphine, while they showed generally lower potencies compared to 14-OMO (**1**) in the acetic-acid-induced writhing assay after s.c. administration to mice (Table 4) [36,42]. Subcutaneous and local intraplantar (i.pl.) administration of 6-Gly-substituted **2a/b** (HS-730/HS-731) and 6-L-Phe conjugates **4a/b** (HS-937/HS-938) also produced antihyperalgesic effects in the formalin test in rats, with increased potencies compared to morphine (Table 4) [40,44]. In rats with neuropathic pain (i.e., sciatic nerve ligation), **2a/b** and **4a/b** compounds were equipotent or somewhat less active in producing antihyperalgesic and antiallodynic effects than morphine after i.pl. injection [44]. In carrageenan-induced inflammatory pain in rats, significant and long-lasting antihyperalgesic actions (up to 4 h) were demonstrated for the 6α- and 6β-Gly-substituted 14-phenylpropoxymorphinans **24a** and **24b**, respectively [37].

Among the developed 6-amino-acid-substituted *N*-methylmorphinans, the 6β-Gly-substituted analogue **3b** (HS-731) was more extensively investigated for its antinociceptive effects in a multitude of diverse pain models, as summarized in Table 5. While it was shown to be very effective as an antinociceptive agent in rodents after systemic parenteral (s.c. and i.p.), central (i.c.v.) and local (i.pl.) application, its significant and prolonged duration of antinociceptive action (up to 4 h) after oral administration to rats with carrageenan-induced inflammatory pain was notable [41]. Furthermore, a recent study described the lack of binding to the human NOP receptor of **3b** (HS-731) [45].

Further pharmacological studies explored the peripheral vs. central components of the antinociceptive effects of 6-amino acid conjugates of 14-OMO (**1**) [40,42]. It was reported that the antinociception of 6β-Gly-substituted **2b** (HS-731) after s.c. administration in the radiant heat tail-flick test in the rat was antagonized by s.c. naloxone methiodide, and not by i.c.v. naloxone, providing evidence that **2b** had a peripheral site of action and not a CNS-dependent mechanism. In contrast, the same s.c. dose of naloxone methiodide did not reverse the antinociceptive effect of morphine after s.c. administration, whereas i.c.v naloxone had antagonized morphine’s effect [40]. In another pain model, the acetic-acid-induced writhing assay in mice, it was also demonstrated the lack of **2b** to enter the CNS, as i.c.v. administration of CTAP, a MOR selective antagonist, did not reverse the antinociceptive effects of systemic s.c. **2b** in mice [42]. These pharmacological data indicate that such compounds produce antinociception via selective activation of peripheral but not central opioid receptors.

The effect of chronic s.c. administration of 6β-Gly-substituted **2b** (HS-731) on the development of antinociceptive tolerance in the radiant heat tail-flick test was recently described [46]. Daily treatment of rats for 14 days resulted in no antinociceptive tolerance for HS-731, indicating that the selective activation of peripheral opioid receptors leads to effective antinociceptive effects without causing antinociceptive tolerance following systemic s.c. administration. Additional behavioral studies remain to establish if other CNS side effects are induced by 6-amino-acid- and 6-dipeptide-substituted *N*-methylmorphinans.

Using the crystal structures of the MOR, DOR, KOR and NOP receptor, the first mechanistic in silico evaluation using molecular docking and molecular dynamics (MD) simulation was reported on the binding mode and interaction mechanisms of **3b** (HS-731) to the opioid receptors [45]. In the same computation study, it was rationalized why **3b** does not bind to the NOP receptor, with the hydroxyl group being likely to abolish ligand binding to the NOP receptor in that it mimics the Tyr residue within the message address of endogenous peptides for the classical opioid receptors instead of the Phe residue within the message address of nociceptin, the NOP receptor agonist [45].

The presence of amino acid residues as ionizable functional groups increases polarity and therefore restricts the ability of the molecule to pass the BBB. The calculated coefficient of distributions at physiological pH, clogD_7.4_, of 6-amino-acid- and 6-dipeptide-substituted *N*-methylmorphinans (**2a**–**22a**), ranging between −5.64 and −0.85 (Table 2 and Table 3), indicated their poor capability to enter the CNS. Such molecules also showed increased hydrophilicity compared to that of morphine, 14-OMO (**1**) and POMO (**23**) (Table 2 and Table 3) [36].

Experimental animal studies intended to explain the limited access to the CNS of opioid analgesics with different physicochemical properties and transport mechanisms by the dose ratio, that is, the ratio of the peripheral and central dose producing a 50% antinociceptive effect (ED_50_) [33,47,48]. Thus, drugs with poor BBB penetration should display a high dose ratio. In this context, in a rat model of acute thermal nociception pain (radiant heat tail-flick test), much higher activity dose ratios of the peripheral (s.c.) vs. central (i.c.v.) antinociceptive potencies (ED_50_) were calculated for the 6-amino acid conjugates **2a/b** (HS-730/HS-731), **3a/b** (HS-935/HS-936), and **4a/b** (HS-937/HS-938)) compared to the ratios of centrally penetrating MOR agonists, morphine, fentanyl and 14-OMO (**1**) (Table 6) [40]. This indicated that the introduction of amino acid residues at position 6 in *N*-methymorphinans is important in limiting penetration into the CNS.

Additional evidence on the peripheral site of action of the 6-amino-acid- and 6-dipeptide-substituted *N*-methymorphinans came from pharmacological antagonist studies [36,37,40,41,42,43]. Antinociceptive effects in different pain models after systemic s.c., i.p. or p.o. administration to rodents were consistently reported to be blocked by naloxone methiodide, a peripheral opioid antagonist, which does not cross the BBB [28]. In formalin-induced inflammatory pain and neuropathic pain induced by sciatic nerve ligation, the antinociception of 6-Gly-substituted **2a/b** (HS-730/HS-731) and 6-L-Phe conjugates **4a/b** (HS-937/HS-938) following local i.pl. injection was also antagonized by naloxone methiodide, demonstrating activation of peripheral opioid receptors [44]. Similar observations were made for others 6-amino-acid- and 6-dipeptide-substituted *N*-methylmorphinans in a model of visceral pain, and the acetic-acid-induced writhing assay, after systemic s.c. administration to mice [36].

#### 2.2.2. 6-*O*-Sulfate Esters of Morphine and Codeine

Further chemical strategies to limit the penetration of morphinans from the periphery to the CNS following systemic administration include 6-*O*-sulfation. Sulfate conjugation generally leads to an increase in the water solubility of the compounds (fully ionized at neutral pH) yet is a biotransformation step in humans to facilitate the excretion of xenobiotics, including opioids [49]. Therefore, 6-*O*-sulfate esters of morphine and codeine were synthesized, such as morphine-6-*O*-sulfate (M6SU) [50,51], codeine-6-*O*-sulfate (C6SU) [50,51,52], 14-methoxymorphine-6-*O*-sulfate (14-*O*-MeM6SU) [53], 14-methoxycodeine-6-*O*-sulfate (14-*O*-MeC6SU) [53] and others [51,54,55].

For the synthesis of sulfate esters, a number of procedures have been reported and reviewed [56,57]. The most common methods are the reaction of phenol or alcohol with chlorosulfonic acid in pyridine, and with trimethylamine-SO_3_ complex or pyridine-SO_3_ complex in DMF, 1,4-dioxane or pyridine. The early synthesis of M6SU and C6SU was based on employing chlorosulfonic acid as a sulfonating reagent [50]. Other synthetical procedures of sulfate esters using pyridine-SO_3_ complex were reported for morphine derivatives [51,55]. For example, preparation of 6-*O*-sulfate esters of morphine and codeine, M6SU and C6SU, respectively, was achieved by means of the sulfation of the C-6 hydroxyl function with pyridine-SO_3_ complex (Figure 5) [51].

Morphine has two hydroxyl groups (C-3 phenolic and C-6) that can be readily sulfated (Figure 5). The reactivity difference of the two hydroxyl groups is not sufficient; however, it is for direct regioselective sulfation to produce the monoester M6SU. In order to obtain M6SU, acetyl-protecting groups at the C-3 phenolic group were used. Selective acetylation of the phenolic hydroxyl moiety was attained upon stirring morphine with acetic anhydride, resulting in compound **25**. The sulfation of the acetylated derivative **25** was achieved by the general method using pyridine-SO_3_, generating **26**, with subsequent alkaline hydrolysis of the protecting group to afford M6SU (Figure 5) [51]. Correspondingly, direct sulfation using pyridine-SO_3_ yielded the 6-*O*-sulfate ester of codeine, C6SU (Figure 5) [51].

The synthesis of the corresponding 14-methoxy analogues of M6SU and C6SU, namely 14-*O*-MeM6SU and 14-*O*-MeC6SU, respectively, was reported (Figure 6) [53,58]. For the preparation of 14-*O*-MeM6SU, 14-OH-codeinone was used for the synthesis of 14-*O*-methyl-codeinone (**27**), which was selectively demethylated in the 3-*O* position. The resulting 14-*O*-methyl-morphinone **28** was reduced by sodium borohydride in methanol, affording **29**, and esterified after selective acetylation of the phenolic 3-OH group. The sulfation of the acetylated derivative **30** was achieved by the general method using pyridine-SO_3_, generating **31**, with subsequent alkaline hydrolysis of the acetyl protecting group to afford 14-*O*-MeM6SU (Figure 6) [58]. The synthesis of 14-*O*-MeC6SU was accomplished by a similar procedure, in which 14-*O*-methyl-codeinone (27) was reduced to **32**, and sulfation using pyridine-SO_3_ yielded the 6-*O*-sulfate ester (Figure 6) [53].

The SAR outcome on the 6-*O*-sulfate substitution in morphine and codeine was reported [53,58]. In vitro binding studies using rodent brain membranes established that the introduction of a 6-*O*-sulfate group in morphine decreased the affinity to the MOR of M6SU in the rat brain by 2-fold, as well as reduced selectivity to the MOR vs. DOR but not vs. KOR (Table 7) [58]. The chemical derivatization of M6SU by introducing a 14-methoxy substituent created 14-*O*-MeM6SU, showing a 10-fold higher MOR affinity than M6SU. In the series of codeine derivatives, the presence of the 6-*O*-sulfate group in C6SU increased the binding affinity to the MOR by 8-fold, whereas a further 29-fold increase was reported for 14-*O*-MeC6SU (Table 7) [53]. In a MVD bioassay, 14-*O*-MeM6SU had higher potency compared to M6SU and morphine in the inhibition of the contraction of the MVD, which was also measured in the [^35^S]GTPγS binding assay using rat brain membranes (Table 8). The same profile was shown by 14-*O*-MeC6SU when compared to C6SU and codeine. The 3-*O*-methyl substitution in M6SU and 14-*O*-MeM6SU resulting in C6SU and 14-*O*-MeC6SU, respectively, reduced both MOR binding affinity and agonist potency in the MVD bioassay and the [^35^S]GTPγS binding assay (Table 7 and Table 8).

Animal studies demonstrated the antinociceptive efficacy of 6-*O*-sulfate-substituted analogues of morphine, M6SU and 14-*O*-MeM6SU, and those of codeine, C6SU and 14-*O*-MeC6SU, in different models of acute nociception, visceral pain, inflammatory pain and neuropathic pain in mice and rats after central (i.c.v. and i.t.) and systemic (s.c., i.p. and p.o.) administration (Table 9) [46,50,52,53,58,59,60,61,62,63,64,65,66]. Generally, the introduction of a 6-*O*-sulfate group in morphine was reported to increase antinociceptive potencies. Additionally, the presence of a 14-methoxy group in 14-*O*-MeM6SU caused a further augmentation in the antinociceptive potency. The high efficacy demonstrated by the 6-*O*-sulfate substituted analogues, M6SU, C6SU, 14-*O*-MeM6SU and 14-*O*-MeC6SU in pathological pain models, including the writhing assay, formalin-induced inflammatory pain, complete Freund’s adjuvant (CFA)-induced inflammatory hyperalgesia and neuropathic pain (chronic construction nerve injury (CCI), and streptozocin (STZ)-induced) following systemic and central administration to rodents (Table 9) is notable. Significant antinociceptive effects were also reported following local i.pl. administration to rats [62,65,67]. The involvement of peripheral opioid receptors to the antinociceptive effects of the 6-*O*-substituted analogues of morphine and codeine was pharmacologically demonstrated based on the reversal of the effects by the peripheral opioid antagonist, naloxone methiodide.

Similar to the 6-amino-acid-substituted derivatives of 14-OMO (**1**) (Table 6), the 6-*O*-sulfate substituted analogues of morphine and codeine, M6SU, 14-*O*-MeM6SU and 14-*O*-MeC6SU, showed a large peripheral vs. central antinociceptive ED_50_ dose ratio compared to morphine and fentanyl in the acute thermal nociception (radiant heat tail-flick test in rats) and visceral pain (acetic acid-induced writhing assay in mice) (Table 10) assays, indicating limited penetration into the CNS.

Pharmacological data that 6-*O*-sulfate substituted analogues of morphine and codeine show reduced potential to induce CNS side effects in animals was reported [60,62,64,65]. First, pharmacologically, limited access to the CNS was established when the antinociceptive effects of a peripherally injected opioid agonist were sensitive to co-administered peripherally acting opioid antagonists, such as naloxone methiodide. This property was observed for M6SU, 14-*O*-MeM6SU, C6SU and 14-*O*-MeC6SU in various pain models [53,62]. However, in these studies, it was clear that the titration of the dose is indispensable to achieve a peripheral antinociceptive effect [46,53,62].

Second evidence that supports the limited CNS penetration of 6-*O*-sulfate substituted morphinans was the reduced sedative effects in animals [60,62,65]. Holtman et al. reported that motor incoordination and hypolocomotion occurred for M6SU at significantly higher i.p. doses (at least 10-fold) than doses required to produce the desirable antinociceptive effects in rodent models of acute thermal nociception, nerve injury-evoked peripheral neuropathy and inflammatory pain following formalin injection [60]. This was also demonstrated by the absence of 6-*O*-sulfate-substituted analogues of morphine to lengthen the righting reflex following their systemic s.c. administration in rats [62,65]. Since opioid analgesics and general anesthetics are known to produce synergistic effects in combination, yet studies on the impact of systemic administration of 14-*O*-MeC6SU have failed to affect the sleeping time induced by inhaled or i.v. anesthetics, this might explain the probability of limited access of 14-*O*-MeC6SU into the CNS [65,68]. Furthermore, M6SU and 14-*O*-MeC6SU in doses prolonging thiobutabarbital-induced sleeping time showed no significant alterations in respiratory parameters compared to saline-treated rats [62]. Evidence of the limited CNS penetration of C6SU is the occurrence of seizures and high incidence of animals death following direct i.c.v. injection into the brain [52,53].

The effects of 6-*O*-sulfate-substituted analogues of morphine and codeine on another typical, undesirable opioid side effect, i.e., gastrointestinal transit were reported [53,60,62]. The activation of peripheral gut opioid receptors, primarily the MOR, is the crucial mechanism involved in opioid-induced constipation [12]. It was described that M6SU had a good separation based on dose (at least 10-fold) between inhibition of gastrointestinal motility and antinociception after i.p. administration in rats [60]. In addition, M6SU also had a more favorable potency ratio for the delay of gastrointestinal transit and antinociception when compared to morphine. However, another study reported that M6SU produces constipation in mice after s.c. administration in antinociceptive doses [62]. In the same study, 14-*O*-MeM6SU also caused inhibition of gastrointestinal transit, but with a 3-fold difference to the antinociceptive dose in the acetic-acid-induced writhing assay. A recent study reported that C6SU showed less gastrointestinal side effects than 14-*O*-MeC6SU, codeine and morphine [53]. 14-*O*-MeC6SU was more effective in inhibiting gastrointestinal transit than C6SU and codeine, and it was similar to that of morphine.

Behavioral studies in animals reported on the lower propensity for the development of tolerance to antinociception of M6SU following chronic administration [60,63,64]. In the radiant heat tail-flick test in rat, antinociceptive tolerance was developed notably slower for M6SU than morphine (25 vs. 10 days) when these drugs were administered i.p. repeatedly at equipotent doses [60]. Similar observations were made in the hot water tail-flick test and streptozotocin (STZ)-induced diabetes in rats, when during 9 days of chronic treatment, tolerance developed to morphine-treated but not to M6SU-treated rats at equianalgesic doses [63,64]. It was also reported that no cross-tolerance exists between M6SU and morphine, withas M6SU inducing antinociception in morphine-dependent diabetic animals [63]. In vitro stability assays using rat and human plasma, rat brain homogenate and simulated gastric and intestinal fluids demonstrated that M6SU (as sodium salt) is highly stable over a 24 h time period and resilient to either enzymatic- or pH-dependent hydrolysis. In addition, M6SU does not hydrolyze to form morphine in a wide variety of physiologically relevant buffers and biological fluids [69].

14-*O*-MeM6SU was also described to produce less antinociceptive tolerance than morphine in the radiant tail-flick test in mice after s.c. administration, although the applied doses were high enough to produce a central effect [70]. No chronic studies on the development of antinociceptive tolerance were reported for other 6-*O*-sulfate-substituted morphinans.

#### 2.2.3. Morphine-6-glucuronide

Peripheral restriction can also be achieved by *O*-glucuronidation at position 6 in the morphinan skeleton. A prominent example is morphine-6-glucuronide (M6G) (Figure 5), the active metabolite of morphine and a potent agonist to the MOR [47,71]. Approximately 10% of morphine is metabolized to M6G [72]. M6G contributes to the clinical analgesic effect of morphine, showing equivalent analgesia but with an improved side effect profile compared to that of morphine [73,74,75].

M6G has been reported to have a binding affinity to the human MOR of 23 nM, and a binding selectivity for the MOR that is 177-fold that of human KOR and 7-fold that of human DOR (reviewed in [76]). Compared to morphine, M6G has been reported to have 6- and 86-fold lower affinity for the human MOR and KOR, respectively, and similar affinity for the human DOR. In other species (i.e., mouse, rat, guinea pig), the M6G:morphine affinity ratio for the MOR ranges from one to four [77].

Animal studies showed from the systemic administration and use of a variety of animal models that M6G is a potent antinociceptive agent (for reviews, see [77,78]). Comparisons of antinociceptive activities of M6G and morphine have been described after systemic (s.c., i.p., i.v. and p.o.) and central (i.c.v. and i.t.) administration to rats and mice. Depending on the experimental model and the species studied, reported relative potencies of M6G to morphine vary from 2:1 to 678:1.

M6G is a very hydrophilic molecule and is therefore expected to have reduced capability to penetrate the BBB [79]. In vitro studies suggest that M6G is a substrate for P-glycoprotein [80,81]. In vivo, however, M6G transport was not affected by P-glycoprotein [82]. In humans, an analgesic effect of M6G was reported in experimental pain models, where i.v. administration of M6G reduced hyperalgesia induced by freeze lesions and excessive muscle contraction [73]. The results of this study indicated that M6G had antihyperalgesic effects in inflammatory pain through the activation of peripheral opioid receptors. The lack of central opioid effects of M6G was established by a lack of change of the pupil size and absence of other opioid-related CNS effects, including nausea, vomiting, itchiness, hiccup and sedation [73]. In clinical studies, the i.v. administration of M6G was found to be as active as morphine over the first 24 h postoperatively [74], and its analgesic efficacy was similar to that of morphine at later time points, although less during the first four postoperative hours [75]. A comprehensive review on the chemistry and preclinical and clinical pharmacology of M6G is beyond the scope of this review, and we recommend extended and topical reviews on this topic [76,77,78,83,84].

### 2.3. Nanocarrier-Based Approaches of Drug Delivery

A promising strategy to alter the pharmacokinetic profile and improve therapeutic effects of drugs is nanotechnology, that is, the use of biocompatible nanocarriers, including nanoparticles, liposomes, nanocapsules, micelles, dendrimers and nanotubes that may carry different therapeutic agents (for reviews, see [85,86,87,88,89,90,91]). The advantage of such nanocarriers is the direct delivery of drugs to the region or cells of interest, improved efficacy and decreased risk of negative side effects. Furthermore, carriers should be biologically stable, should protect the drug from degradation and the host body from toxic side effects, and should be able to deliver the loaded drug specifically to the target cell population in vivo (for reviews, see [85,86,87,88,89,90,91]). Nanotechnology has been extensively examined for tumor-directed delivery of chemotherapeutics to reduce their off-target toxicity [89], and has also been proposed for pain management [90,91].

Recently, a nanocarrier-based approach was developed that uses hyperbranched, dendritic polyglycerols (PG) to selectively deliver morphine to peripheral inflamed tissue [92]. Morphine was covalently bound to PG, via a cleavable ester linker sensitive to esterases and low pH. The rationale was that due to its high molecular weight and hydrophilicity, the i.v. injection of PG-morphine will not cross the BBB, but will selectively extravasate from leaky blood vessels characteristic of inflamed tissue. The local low pH and leukocyte esterases will then trigger the release of morphine from PG-morphine to reduce pain behavior [92]. In radioligand binding studies using human embryonic kidney (HEK) 293 cells stably expressing the rat MOR, PG-morphine was 10,000 times less effective than morphine (IC_50_ of 18.2 µM vs. 0.002 µM, respectively), indicating that PG-M does not bind to the MOR [92]. Different to morphine, dosages containing equivalent amounts of morphine-free PG-morphine exclusively activated peripheral opioid receptors following systemic i.v. administration in rats with unilateral hind paw inflammation. PG-M selectively induced antinociception in the inflamed paw in a naloxone-methiodide-dependent manner. Free morphine was only detected in inflamed paw tissue, but not in the contralateral, non-inflamed paw tissue, blood and brain of rats [92]. Furthermore, PG-M was reported not to cause sedation and constipation at antinociceptive doses after i.v. injection in rats with inflammatory pain [92]. However, the organ toxicity and broader side effect profile, including abuse potential and effects on respiration of PG-M were not reported yet to strengthen the clinical applicability of this strategy, which is able to deliver morphine exclusively in injured tissue, precluding not only CNS side effects but also constipation.

Further, morphine-loaded hydrogels have been reported. Preclinical studies demonstrated that peptide-based hydrogels loaded with morphine as new controlled-drug delivery systems produced effective, sustained antinociceptive effects in mice after s.c. administration, and no sedative effects were observed [93,94].

## 3. Conclusions

In the current context of the ‘opioid crisis’, the development of new opioid analgesics with improved pharmacology (i.e., efficacy in various pain conditions and reduced capability of inducing unwanted side effects) is of crucial clinical and public health attention. Diverse opioid analgesic discovery efforts are therefore made towards identifying effective and well-tolerated opioids that have an improved benefit/risk ratio compared with currently available drugs.

The basis for drug discovery targeting peripheral opioid receptors is supported by the knowledge that opioid receptors are expressed in the CNS, PNS and peripheral tissues. Furthermore, activating opioid receptors in the periphery leads to an effective analgesic response, and the most serious opioid-related adverse effects (i.e., apnea, sedation, physical dependence and addiction) are due to the activation of opioid receptors in the CNS. Thus, peripherally restricted opioids are viewed as viable targets to avoid many of the lethal side effects associated with opioids targeting the CNS.

In this review, we discussed different peripheralization strategies applied to opioids. Emphasis was placed on the morphinan class of opioid ligands represented by morphine and its structurally related analogues that are used extensively not only clinically but also as experimental tools and that are important as scaffolds for the design of new ligands. Broad chemical and pharmacological work was performed on modifications of the morphinan scaffold to reduce the ability to cross the BBB, and substantial achievements have been made in the field, increasing the feasibility for clinical application. We discussed chemical variations on the morphinan skeleton to increase the hydrophilicity, such as quaternization of the morphinan nitrogen (N17), the introduction of polar/ionizable substituents at C-6 position (i.e., amino acid, sulfation and glucuronidation), and nanocarrier-based approaches to selectively deliver morphine to peripheral tissue. Although the available preclinical and clinical data are favorable to the potential use of these compounds, their clinical impact, and the extent to which they will replace existing opioids, needs further investigations.

## Data Availability

Not applicable.

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
