# Peer review of "Peripheralization Strategies Applied to Morphinans and Implications for Improved Treatment of Pain"

_molecules, 2023, doi:10.3390/molecules28124761_

Round 1
Reviewer 1 Report
The reviewer would like to declare no conflict of interest with the authors and their afffiliations.
Generally, the review article by Schmidhammer and colleagues is very well written. However, some minor comments has to be address before the manuscript can be accepted.
1. The authors repeatedly used the phrase "global opioid crisis". Actually, the opioid crisis, or opioid epidemic, was declared by the US-FDA, and reports showed that only the North America and some parts of the European Continents face opioid crisis. In contrary, patients in some countries have limited access to opioids... It is highly recommended that the authors include 1 or 2 paragraph describring the epidermiology of opioid epidemic in the introduction section.
2. Peripherally-restricted opioids receptor agonists as potential pain medication can be a viable strategy to address the central side effects of CNS-active opioids, however, BBB-impermeable opioid receptor agonist has been in clinical use for many decades, ie loperamide for GI motility. The authors should compare the efficacies and potential applications of the proposed novel candidates in the review article, with those are already applied in clinical setting.
Author Response
Reviewer 1
Generally, the review article by Schmidhammer and colleagues is very well written. However, some minor comments has to be address before the manuscript can be accepted.
Author’s reply: We would like to thank the Reviewer for reviewing our manuscript and the constructive comments. We have given careful consideration to all issues raised by you as indicated in the point-by-point response. Changes are highlighted in the revised manuscript.
- The authors repeatedly used the phrase "global opioid crisis". Actually, the opioid crisis, or opioid epidemic, was declared by the US-FDA, and reports showed that only the North America and some parts of the European Continents face opioid crisis. In contrary, patients in some countries have limited access to opioids... It is highly recommended that the authors include 1 or 2 paragraph describring the epidermiology of opioid epidemic in the introduction section.
Author’s reply: We wish to thank the Reviewer for this comment. We have made amendments in the Introduction, though a more extensive discussion of the epidemiology of opioid epidemic is beyond the scope of this review manuscript (see lines 42-46).
- Peripherally-restricted opioids receptor agonists as potential pain medication can be a viable strategy to address the central side effects of CNS-active opioids, however, BBB-impermeable opioid receptor agonist has been in clinical use for many decades, ie loperamide for GI motility. The authors should compare the efficacies and potential applications of the proposed novel candidates in the review article, with those are already applied in clinical setting.
Author’s reply: We wish to thank the Reviewer for this comment.
The reviewer refers to loperamide, which is the only peripheral opioid agonist using clinically as anti-diarrheal drug and not as analgesic drug. In our review manuscript, we have paid attention and have made reference to preclinical and clinical studies where the new peripherally-restricted opioid agonists were compared for their analgesic efficacy and/or side effects to clinically used opioids (i.e. morphine, codeine, fentanyl) (see Sections 2.2.1, 2.2.2., 2.2.3, and 2.3., and Tables 3, 5, 8 and 9).

Reviewer 2 Report
The authors present a detailed review on the strategies being implemented on peripherally acting (and not CNS acting) opioids as a means to obtain potential analgesic effects without the unwanted and most dangerous side-effects of opioids. The review is very well written and well structured so that it is easy to follow. I find that there is very little wrong with this review. My only very minor comment is that I believe the drug delivery section should be marginally expanded with additional relevant references about the nuanced benefits of localized delivery and sustained release of the nanoparticle systems. doi.org/10.1002/jps.24022 could also be an interesting additional reference.
Author Response
Reviewer 2
The authors present a detailed review on the strategies being implemented on peripherally acting (and not CNS acting) opioids as a means to obtain potential analgesic effects without the unwanted and most dangerous side-effects of opioids. The review is very well written and well structured so that it is easy to follow. I find that there is very little wrong with this review.
Author’s reply: We would like to thank the Reviewer for reviewing our manuscript and the constructive comments. We have given careful consideration to all issues raised by you as indicated in the point-by-point response. Changes are highlighted in the revised manuscript.
My only very minor comment is that I believe the drug delivery section should be marginally expanded with additional relevant references about the nuanced benefits of localized delivery and sustained release of the nanoparticle systems. doi.org/10.1002/jps.24022 could also be an interesting additional reference.
Author’s reply: We wish to thank the Reviewer for this comment and the recommended reference. While the recommended reference targets CNS delivery of opioids (with fentanyl as example), the focus of our review is on peripheralization strategies applied specifically to opioid morphinans (as the majority of clinically used opioid analgesics belong to the class of morphinans) to restrict their BBB permeability thus minimizing central exposure and the associated undesired side effects.
An extensive review on localized delivery and sustained release of the nanoparticle systems and their benefits is beyond the scope of this review article. However, we have cited several descriptive reviews in the field and briefly referred to the advantages of nanoparticle drug delivery systems (see pages 568-573 and references 85-91).
